# Comparing Intuitions about Agents' Goals, Preferences and Actions in Human Infants and Video Transformers

**Alice Hein and Klaus Diepold**
Chair of Data Processing
TUM School of Computation, Information and Technology
Technical University of Munich, Germany
{alice.hein, kldi}@tum.de

## Abstract

Although AI has made large strides in recent years, state-of-the-art models still largely lack core components of social cognition which emerge early on in infant development. The Baby Intuitions Benchmark was explicitly designed to compare these "commonsense psychology" abilities in humans and machines. Recurrent neural network-based models previously applied to this dataset have been shown to not capture the desired knowledge. We here apply a different class of deep learning-based model, namely a video transformer, and show that it quantitatively more closely matches infant intuitions. However, qualitative error analyses show that model is prone to exploiting particularities of the training data for its decisions.

## 1 Introduction

The foundations of "commonsense psychology" emerge early on in a human's development: Even pre-verbal infants have expectations about agents' goals, preferences and actions [17]. Although deep learning (DL) has made tremendous progress in recent years, this core component of human cognition is still lacking in many state-of-the-art DL models [13]. When tested on the Baby Intuitions Benchmark (BIB), a dataset designed to compare the social cognitive abilities of infants and machines, behavioral cloning (BC) and video prediction models based on recurrent neural networks (RNNs) failed to show infant-like reasoning [6]. We here evaluate a different class of DL model, namely a video transformer (VT), on the BIB dataset.

Recent years have seen the rise of transformers in various areas of AI, including tasks adjacent to social cognition, such as trajectory prediction for cars or pedestrians [20, 15, 2, 18, 9, 19] and spatial goal navigation [4, 1, 5]. As the transformer attention mechanism is based on computing pairwise interactions [14], this family of models constitutes a promising approach for capturing the relations between, e.g., agents and goals in the BIB dataset. However, transformer-based video prediction models require many costly pairwise computations. They are usually trained and evaluated on datasets like *Kinetics-400* [12] or UCF101 [16], where video clip lengths range from 7 to 10 seconds – much shorter than those used in BIB, which may be up to 2 minutes long. We therefore implement some modifications to allow a VT to process BIB episodes, and evaluate the resulting model.

We find that the VT quantitatively more closely matches infant intuitions about agent's goal preferences and efficient actions than previously tested DL baselines. However, qualitative error analyses show that the model fails to generalize systematically on some of the test tasks when agent or environment dynamics differ slightly from background training observations.

4th Workshop on Shared Visual Representations in Human and Machine Visual Intelligence (SVRHM) at the Neural Information Processing Systems (NeurIPS) conference 2022. New Orleans.

Table 1: Overview of BIB tasks.

| | Familiarization trials | Test trial | Expected outcome | Unexpected outcome |
|---|---|---|---|---|
| **Preference** | | Identical to a familiarization trial, but object positions are switched | Agent moves to preferred object at new location | Agent moves to nonpreferred object at familiar location |
| **Multi-agent** | Agent consistently chooses one of two goal objects and moves to it efficiently | New agent appears | New agent moves to object not preferred by familiar agent | Familiar agent moves to previously not preferred object |
| **Inaccessible goal** | | Preferred goal becomes inaccessible | Agent moves to other goal | Agent moves to other goal, even though both are accessible |
| **Efficient agent** | Agent moves efficiently around a barrier towards goal | Barrier is removed | Agent moves efficiently | Agent moves inefficiently |
| **Irrational agent** | One agent moves efficiently, one moves inefficiently | Both agents move inefficiently | Previously inefficient agent moves inefficiently | Previously efficient agent moves inefficiently |
| **Instrumental action** | Agent removes a green barrier (inserts key into lock), then moves to goal | Green barrier gone or inconsequential | Agent moves directly to goal | Agent still moves to key |

## 2 Baby Intuitions Benchmark

BIB is a dataset designed to test whether machine learning systems can discern the goals, preferences, and actions of others [6]. It consists of videos in the style of Heider and Simmel's animations [10], where agents, represented by simple shapes, carry out actions in a 2D grid world. BIB follows the violation-of-expectation (VoE) paradigm, i.e., each video has a familiarization and a test phase. The familiarization phase consists of eight successive trials during which an agent consistently displays a certain behavior, allowing the observer to form an expectation of future actions. The test phase includes an expected outcome (perceptually similar to the previous trials, but involves a violation of expectation), and an unexpected outcome (perceptually less similar, but conceptually more plausible). BIB contains six types of tests tasks, outlined in Table 1. It also contains background training episodes, which share the same structure as the test set. However, only expected trials are provided, and only isolated tasks are trained, such that the systematic combination of acquired knowledge is needed to generalize to the test tasks. For examples and more details on BIB, see A.

Because BIB adopts its tasks and paradigm from developmental cognitive science and provides sufficient data to train DL-based models, it allows for the direct comparison of human and machine performance [6]. A critical first step in this direction was taken by Stojnic et al. [17], who collected infants' responses on a representative selection of BIB episodes and compared them with three state-of-the-art DL models from two classes: Behavioral cloning (BC) and video modeling. Recently, Zhi-Xuan et al. [21] proposed a principled alternative to DL approaches, based on a hierarchically Bayesian Theory of Mind (HBToM). Results from both works serve as comparisons in this paper. Note, however, that HBToM requires access to symbolic states and is specifically engineered to solve BIB-like social cognition tasks, whereas the data-driven baselines and VT model have weaker inductive biases in this regard.

## 3 Methods

Our model consists of a convolutional neural network (CNN) encoder, a transformer component, a CNN decoder, and a feedforward output layer. A schematic visualization is shown in Figure 1. The CNN encoder has two convolutional layers and two max-pooling layers. For each $3 \times 84 \times 84$ input image, it produces a $30 \times 21 \times 21$ representation, which we concatenate with x- and y-position encodings, yielding $32 \times 21 \times 21$ image patches. As attending over every pixel would be computationally prohibitive, the CNN encoder was designed to reduce the frame's resolution by extracting higher-level features, while retaining a sufficient level of spatial detail.

The transformer component consists of three standard five-layer attention blocks with 8 heads of input dimension 32 and hidden dimension 256. The number of heads and layers was chosen to strike a balance between performance and computational complexity. The first block performs cross-attention over the test trial's encoded first frame and the previous familiarization trials, effectively "priming" the model by calculating the influence of previous observations on the current input. Because attending over every patch, frame, and trial would be extremely computationally expensive, we only feed in the top-$k$ patches per frame that display the highest change compared to the previous frame. $K$ was set to 3, as using a higher number would have exceeded the memory resources in our training setup. The results of attending over each trial are then averaged and passed through a self-attention

block, followed by another cross-attention block. This block attends over past steps in the test trial, encoded in the same way as the familiarization trial frames. Intuitively, the second attention block serves to compute a global trajectory plan, whereas the third attention block calculates the agent's next move based on its actions so far. In a final step, the outputs of the transformer component are passed through a linear layer, which produces a $1 \times 21 \times 21$ prediction of the agent's next position, and a CNN decoder, which produces a $3 \times 83 \times 84$ prediction of the video's next frame.

As in Gandhi et al. [6], the videos' frame rate was downsampled by a factor of 5. We used a maximum sequence length of 90. Frame rates of longer sequences were interpolated to fit the maximum length. Of the BIB background episodes, we used 80% for training, 15% for testing, and 5% for validation. Models were trained using the Adamax optimizer for a total of 6 epochs. The batch size was set to 6 because of the VT's high memory requirements. We tested the models on the validation set in five evenly spaced intervals per epoch and saved the model with the lowest validation loss to avoid overfitting. Our loss function consisted of the sum of two terms. The first term was the binary cross-entropy (BCE) loss between the prediction of the agent's next step and the actual agent position. To address the imbalance between the "agent" and "no-agent" class, we employed a weighted version of the BCE loss, which is widely used in instance segmentation [11]. The second term was the mean squared error (MSE) between the prediction of the next frame and the actual next frame, upweighted by a constant factor so that both loss terms were scaled evenly. This second term was introduced because transformers may disregard agent identities unless incentivized otherwise [20]. For tasks like *preference*, which relies on the preservation of agent shapes and colors, we therefore found it improved performance to include an auxiliary reconstruction loss. During evaluation, only the main BCE loss was used. On a 16-Core AMD EPYC 7282 server with six GeForce RTX 2080 GPUs, training time was around 3 hours per epoch. Our code is available at github.com/zero-k1/BIB-VT.

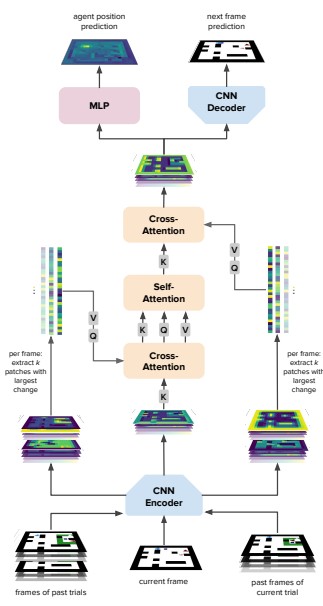

Figure 1: Schematic visualization of the VT architecture.

## 4 Results

In total, we trained five models with different random seeds, and we report their average performance and standard deviations. The baseline DL models previously tested on BIB used the prediction error of the frame with the highest loss as their metric of "surprise", as this provided better results compared to the mean error over entire trials [6]. In our case, the mean error yielded a higher performance on most tasks, which is why we here report both metrics. Performance comparisons with models previously tested on BIB are shown in Table 2. However, binary VoE accuracies include no information about the magnitude of the difference in surprisal scores between expected and unexpected trials. We therefore also show z-scored means of both the models' average prediction error and infants' looking times, as reported by Stojnic et al. [17], in Figure 2.

### 4.1 Goal-directed

#### 4.1.1 Preference

In contrast to previous DL-based models, the VT seems, at least to some degree, to associate agents with certain goal preferences in the *preference* task (see Figure 2a). To investigate which parts of the familiarization trials the model relied on most for its decisions, we performed a form of occlusion analysis. We used only one trial as the familiarization input (performance was almost identical when using one vs. the full eight trials), and dropped each of the patches fed into the first transformer block in turn. For each patch, we recorded the z-scored difference in prediction error between the expected and unexpected outcome. An example result is shown in Figure 3. Models tended to either focus on

Table 2: VoE Accuracy on BIB tasks. VoE Accuracy denotes whether model error is higher on expected trials than unexpected trials. VT (Mean) uses the avg. error over all test trial frames as the "surprise" metric, whereas VT (Max) uses the error for the frame with the highest loss. Baselines and Video Transformers are data-driven computer vision models, whereas HBToM uses a principled Bayesian solution that requires access to symbolic states. Chance level accuracy 50%

| Task | HBToM | Baselines | | | Video Transformer (ours) | |
|---|---|---|---|---|---|---|
| | | BC-MLP | BC-RNN | Video-RNN | VT (Mean) | VT (Max) |
| Goal-directed | | | | | | |
| *Preference* | 99.7 | 26.3 | 48.3 | 47.6 | 82.1 ± 0.0 | 80.8 ± 0.0 |
| *Multi-agent* | 99.2 | 48.7 | 48.2 | 50.3 | 49.1 ± 0.0 | 49.2 ± 0.0 |
| *Inaccessible goal* | 99.7 | 76.9 | 81.6 | 74.0 | 89.8 ± 0.0 | 85.5 ± 0.0 |
| Efficiency | | | | | | |
| *Efficient agent* | 95.8 | 96.0 | 95.3 | 99.5 | 98.3 ± 0.0 | 98.4 ± 0.0 |
| *Irrational agent* | 96.6 | 73.8 | 56.5 | 50.1 | 29.5 ± 0.1 | 34.1 ± 0.1 |
| Instrumental actions | | | | | | |
| *Instrumental action* | 98.5 | 67.0 | 77.9 | 79.9 | 92.6 ± 0.0 | 84.7 ± 0.0 |

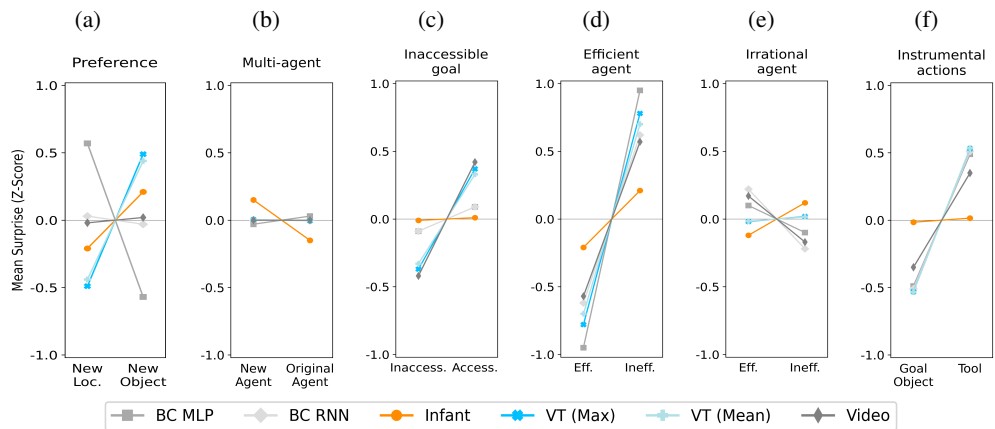

Figure 2: Z-scored means of the models' average surprisal scores and infants' looking times to the expected and unexpected outcomes in the BIB test episodes.

the agent's last or first step. Averaged over all models and episodes, the patch with the largest impact on the final prediction was part of the last two frames of the familiarization trial in 52.6% of cases.

### 4.1.2  Multi-Agent

Similar to the other DL models, the VT does not acquire the desired knowledge from the *multi-agent* background training tasks, which feature both agents moving towards the same single goal across trials. Note that the infants tested on BIB were in fact more surprised at the supposedly "expected" trials (see Figure 2b). Stojnic et al. [17] hypothesize that this may be because of the increased novelty of the new agent. A closer look at the frame predictions produced by the VTs hints at some confusion regarding the agents' identity: In some cases, the model reconstructs the familiar agent in the unexpected trial, rather than the new agent present in the input (see Figure 4 for an example). Averaged over all models and episodes, this was the case 27.9% of the time.

### 4.1.3  Inaccessible

In the *inaccessible-goal* task, the VT model achieves a higher accuracy than previous DL models. It exhibits a stronger deviation in surprise than the infants, who were indifferent on this task (see Figure 2c). Stojnic et al. [17] posit that infants may have considered the new barrier in the expected outcome as indicative of a new environment and not carried over any goal preference expectations from the familiarization trials. Although the VT has a lower prediction loss on the expected outcome in most cases, it is more "split" than in the single-object case (see Figure 5 for an example prediction). Averaged over all models and episodes, the entropy of the models' prediction on the test trial's last

(a)      (b)      (c)

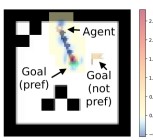

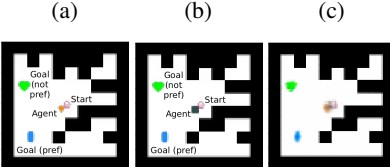

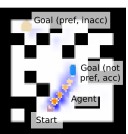

Figure 3: Z-scored impact of omitting a patch from the *preference* familiarization trial.

Figure 4: 3a: Unexpected *multi-agent* outcome (familiar agent). 3b: Expected outcome (new agent). 3c: Prediction for expected outcome.

Figure 5: *Inaccessible goal* task. Predicted agent positions marked blue.

frame was 1.10 for the expected, and 1.47 for the unexpected outcome. For comparison, the average entropy for the last frame of the single-object *efficiency-time* trials was only 0.58.

## 4.2 Efficiency

Similar to previous models, the VT's VoE accuracy on the *path-control* and *time-control* tasks are nearly perfect – the model strongly expects agents to move towards their goal efficiently. This is in accordance with infant's intuitions (see Figure 2d). On the *inefficient-agent* task, the VT tends to be more surprised at the previously inefficient model moving inefficiently than at the previously efficient agent doing so. Although not necessarily a desired outcome, this is actually more in line with the intuitions of the infants tested on BIB, who attributed rational action both to previously efficient and inefficient agents in a new environment (see Figure 2e). When we compare the impact of the familiarization trials featuring the efficient vs. inefficient agent

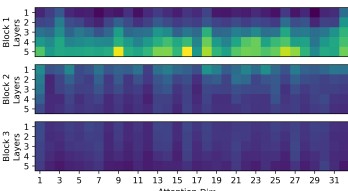

Figure 6: Avg. difference in the VT layers' activations when processing the episodes' unexpected vs. expected familiarization trials, featuring a rational or an irrational agent, respectively.

on the VT model (see Figure 6), we see that a similar mechanism is at work: The lowest levels, which attend over past familiarization trials, show differences in activation. However, these differences all but disappear throughout the higher layers. This leads to the inefficient agent being treated in the same way as the efficient one, which explains the mean surprise score being almost the same in both cases. The slightly larger error for the inefficient agent most likely stems from the fact that irrational agents are not seen during training, leading to higher prediction uncertainty.

## 4.3 Instrumental Actions

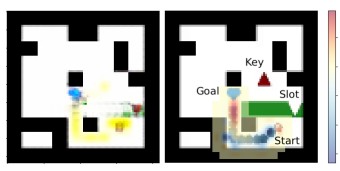

(a) Predicted last frame and agent trajectory patch (yellow).

(b) Z-scored impact of each test trial on final MSE error.

Figure 7: Prediction on an *instrumental-action* task.

Compared with the other DL models, the VT performs similar on episodes with no barrier, and better on episodes with inconsequential or blocking barriers. Again, infants were indifferent on this task (see Figure 2f). Stojnic et al. [17] note that they may have failed to recognize the instrumental actions because they were causally opaque. Although the VT is correct in most cases in terms of VoE accuracy, it, too, seems to not have quite understood the causal mechanism. A look at the frame predictions shows that the model usually expects the disappearance of the key on the first step, even though the agent has not collected and inserted it. Averaged over all models and episodes, the VT at least partly predicts the key's position as the agent's first step in 47% of cases, even though the key is mostly far away from the agent. This is most likely because the key is always right next to the agent in the background *instrumental-action* tasks, and thus constitutes its first step. The VT also often predicts the disappearance of the green barrier towards the end of the episode, even though the key was not inserted. This is most likely because the green barrier has always disappeared by the time the agent reaches the goal in the background tasks. Occlusion analyses support this hypothesis: The parts of the test trial that most contribute to the z-scored MSE prediction error on expected *instrumental-action* outcomes were usually the agent's first and last steps (see Figure 7 for an example).

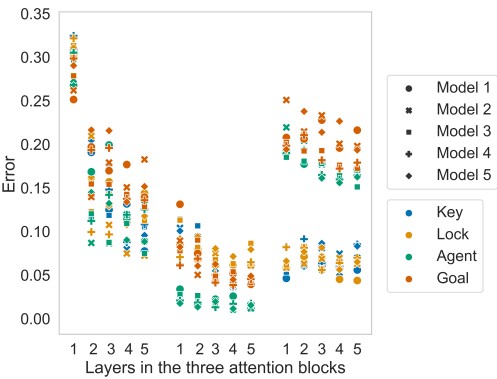

Figure 8: Weighted binary cross-entropy loss for linear regression models trained on decoding the current position of goals, agents, and sub-goals, from each layer of the five models.

## 4.4 Decoding experiment

Inspired by probing analyses of pre-trained language models [3], we train linear regression models to predict the current position of the agent, goal, and sub-goals (keys and locks), based on the concatenated output of the eight attention heads in each layer of each video transformer block. We use the background training set for optimization and display the results for the background validation set in Figure 8. In general, we see errors decrease in the deeper layers of the attention blocks, indicating more focused attention heads. The heads in the first block, which attends over familiarization trials, do not display a large degree of specialization regarding the analysed categories. However, at least in the higher layers, the agent, key, and lock categories have a comparatively lower decoding error than the goal category. Note that the agent's position often corresponds to the key and lock position for long stretches of instrumental action trials, as the agent waits for the green barrier to disappear after having inserted the key into the lock. The second block, which self-attends over the test trial's first frame, has the lowest decoding error across categories and a particular low error for the agent's current position. The third block shows a clear separation between categories, with locks and keys displaying a much lower decoding error than goals and agents. This is presumably because the third attention block autoregressively predicts the agent's next step, which, as mentioned, often coincides with the key and lock position while the agent is waiting in place for the barrier to disappear. In summary, the video transformer seems to have learned to implicitly keep track of relevant semantic categories, such as agents, goals, and subgoals, which are usually modelled as explicit variables in Bayesian approaches.

## 5 Discussion and Conclusion

In conclusion, the VT model tested in this paper outperforms previous DL-based baselines on the *preference*, *inaccessible-goal*, and *instrumental-actions* BIB tasks in terms of VoE accuracy. Its surprisal scores are also more in line with infants' expectations than previous DL models, in that it tends to represent agents' actions as directed towards goals, rather than locations, and defaults to expecting rational actions. This suggests that the transformer's attention mechanism can be helpful in acquiring intuitions about agents' goals, preferences, and actions, purely from predicting the next step in videos. However, a qualitative analysis of the VT's errors also demonstrated the pitfalls of this approach: Models may exploit the particularities of a training dataset in an unintended way [7, 8], e.g. by associating the disappearance of the green barrier in the *instrumental-actions* task with the agent's first and last step rather than the key mechanism. This may be mitigated with a more realistic data setting, where models can gain experience with diverse agents and disambiguate causes and effects of instrumental mechanisms interactively, in a manner closer to human infants. The findings also support the benefit of investigating hybrid architectures that incorporate methods which explicitly model human intuitions, such as HBToM, to take advantage of both the flexibility of DL-based approaches and the data efficiency and robustness of principled Bayesian models.

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

# A Appendix

## A.1 BIB Tasks

### A.1.1 Goal-directed actions

The *preference* task (1,000 episodes) tests whether an observer represents agents as having a preference for goal objects, rather than locations. The setup consists of two goals and an agent, whose starting position is fixed. In the familiarization trials, the agent consistently moves towards the same object. Goal locations and identities are correlated, such that preferred and nonpreferred goals have a similar position across trials. In the test phase, the two objects appear in positions previously seen during familiarization. However, goal identities are switched. In the expected outcome, the agent moves to the preferred object. In the unexpected trial, the agent follows the same trajectory as seen during familiarization and moves to the nonpreferred object (see Figure 9a).

The *multi-agent* task (1,000 episodes) tests whether an observer attributes specific goal preferences to specific agents. The setup consists of two goal objects appearing at different positions across trials, and an agent with a fixed starting position. Again, the agent moves repeatedly to the same object during familiarization. In the unexpected test outcome, the agent moves towards its nonpreferred goal. In the expected outcome, a new agent replaces the previously seen one and moves toward the familiar agent's nonpreferred object. The unfamiliar agent choosing a new goal should be less surprising than a familiar agent switching preference (see Figure 9b).

The *inaccessible-goal* task (1,000 episodes) tests whether an observer understands the principle of solidity, and that physical obstacles may restrict agents' actions. The familiarization trials are identical to the *multi-agent* task. In the expected test trial, the previously preferred object is made inaccessible by a black barrier, and the agent moves to the other goal. In the expected test trial, the agent switches goal preference despite both objects staying accessible (see Figure 9c).

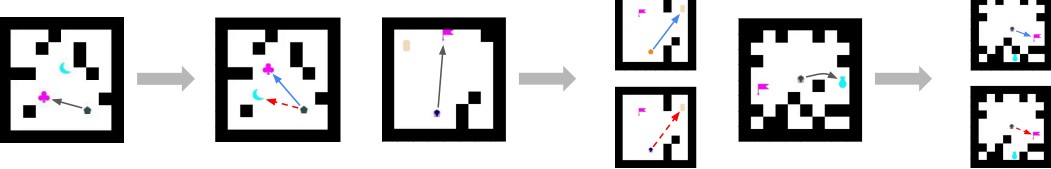

(a) Example of a *preference* task. Goal locations are switched for testing. In the expected outcome, the agent still chooses the same object. In the unexpected outcome, the agent instead follows the familiar path to its nonpreferred goal.

(b) Example of a *multi-agent* task. A new agent appears in the test trial. This new agent choosing the other agent's nonpreferred object (top right) should be less surprising than the familiar agent doing so (bottom right).

(c) Example of an *inaccessible-goal* task. The agent switches goals in the test trial. This should be expected if the preferred object is inaccessible (top right), but unexpected if both objects are accessible (bottom right).

Figure 9: Examples of goal-directed action tasks. Agents move repeatedly to the same goal during familiarization (left), while test trials differ by task type (right). Blue solid lines represent expected outcomes, red dashed lines represent unexpected outcomes.

### A.1.2 Efficient actions

The *efficient-agent* task tests whether an observer expects agents to move efficiently towards their goal. It consists of two subtasks: path control (1,500 episodes) and time control (1,000 episodes). In both subtasks, the setup consists of one goal object and one agent. During familiarization, the agent moves efficiently towards the object, but must navigate around a barrier to reach it. This obstacle is removed in the test phase. In both subtasks, the expected outcome consists of the agent moving efficiently towards its now-unobstructed goal. For the path control task, a previously seen combination of agent and goal location is used, and the unexpected outcome consists of the agent moving along the familiar, but now inefficient, trajectory (see Figure 10a). For the time control subtask, the goal object is placed closer to the agent and the unexpected outcome consists of the agent following a path that is inefficient, but takes up the same amount of time as the efficient one.

The *inefficient-agent* task (890 episodes) tests whether an observer forms expectations about the actions of irrational agents. During familiarization, an agent is shown either moving efficiently, as in the *efficient-agent* task, or inefficiently. In the test phase, the agent is shown moving inefficiently to the goal. This should be an unexpected outcome if the agent previously behaved rationally, and an expected outcome if the agent previously behaved irrationally (see Figure 10b).

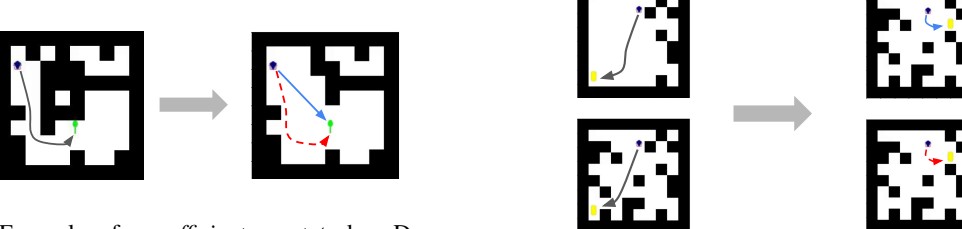

(a) Example of an *efficient-agent* task. During familiarization, the agent navigates efficiently around an obstacle to reach its goal. The barrier is removed during testing. The agent is expected to now move efficiently, rather than following the same path as before.

(b) Example of an *inefficient-agent* task. An agent that moves inefficiently during familiarization (top) is expected to continue doing so during testing, whereas an efficient agent (bottom) beginning to move inefficiently should be surprising.

Figure 10: Examples of efficiency tasks. Familiarization trial shown on the left, test trials on the right. Blue solid lines represent expected outcomes, red dashed lines represent unexpected outcomes.

### A.1.3 Instrumental actions

The *instrumental-action* task (987 episodes) tests whether an observer can recognize an agent's action sequences as instrumental and directed towards higher-order goals. The setup consists of a goal, an agent, a removable green barrier with a lock, and a key, represented by a red triangle. During familiarization, the goal is obstructed by the green barrier. The agent collects the key, inserts it into the lock, removes the barrier, and moves to the goal. In the test phase, a key is still present, but the green barrier is either absent or no longer blocking the goal. In the expected outcome, the agent moves directly towards the goal, whereas it still moves towards the now-obsolete key in the unexpected outcome (see Figure 11).

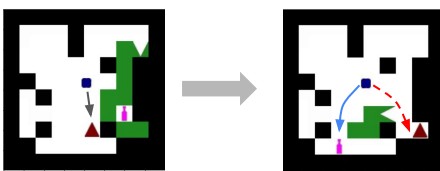

Figure 11: Example of an *instrumental-action* task.

### A.1.4 Background training episodes

To facilitate the training of machine learning models, BIB includes a large number of background episodes which share the same structure, agents, and goal objects as the test set. However, only expected trials are provided during training. The training set is split into four tasks. In order to generalize systematically to the test trials, the model needs to combine knowledge acquired from all four training tasks. In the *single-object* task (10,000 episodes), an agent navigates efficiently to a goal object (see Figure 12a). In the *preference* task (10,000 episodes), the agent consistently chooses one object over another across trials (see Figure 12b). In contrast to the *preference* test task, both objects are located very close to the agent, so navigation is not trained. In the *multi-agent* task (4,000 episodes), the agent moves to a very close-by single goal object (see Figure 12c). At some point during the episode, the agent is replaced with a new agent. This differs from the *multi-agent* test task, where there are two goals which are placed farther away and the new agent only appears in the test trial. In the *instrumental-action* task (4,000 episodes), the agent is initially confined by a green barrier, which it removes with a key in order to move to its goal. This differs from the *instrumental-action* test task in that the barrier surrounds the agent, rather than the goal.

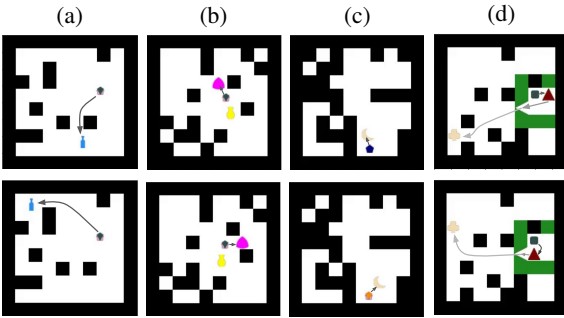

Figure 12: Examples of training trials, consisting of *single-object* (12a), *preference* (12b), *multi-agent* (12c), and *instrumental* (12d) tasks.

