# OpenReview forum: "Comparing Intuitions about Agents’ Goals, Preferences and Actions in Human Infants and Video Transformers"
_NeurIPS.cc/2022/Workshop/SVRHM — SVRHM Poster_

### Official Review · Reviewer_ubwT · 2022-10-07
**Interesting and well written paper. Adding some intuitive explanation of VT successes and failures would further improve the paper**

**Rating:** 9
**Confidence:** 4

**Review:**

This is an exciting paper comparing the performance of visual transformers to infant violation of expectation on a range of social visual tasks. Overall, I thought the paper was very comprehensive and well written. The technical details in particular were very clear.

I think it would help to provide some intuition about *why* VTs were tested in the intro (why might they improve performance on the social tasks mentioned), and especially why they improve performance over other DL models on certain tasks but not others.

A smaller point is that it would also help if there were one-one alignment between the tasks in Table 1 and Table 2. It wasn't clear to me why they were broken into different tasks/sub-tasks in Table 2 and this made it somewhat harder to understand task differences/model performance.

---

### Official Review · Reviewer_yidQ · 2022-10-10
**A powerful model for social goal understanding, but low performance in multi-agent tasks need to be investigated**

**Rating:** 8
**Confidence:** 4

**Review:**

The authors develop a new deep learning based model that is motivated by "commonsense psychology". They evaluate a number of baseline methods for trials of BIB and show that these methods do not perform well on BIB (except HBToM). This suggests that their video transformers will be useful for detecting and quantifying advances in this domain.

I believe that this represents a valuable contribution toward the development of AI that is able to mimic humans' intuitive goal and action understanding. I only have some concerns about the low performance for multi-agent task and blocking barrier task (for VT max values). Interestingly, infants' performances for these two domain have different patterns (figure 7). For multi-agent tasks, infants looked at unexpected outcome less than expected outcome, which shows that they suprised at expected outcome. For the instrumental action, infants had no preference. This was mentioned at page 3 and 4. Checking both human performance and VT models performances with variety of the same scenes for these two areas can be beneficial. I also wondered why HBToM outperformed for every task, but not VT models.

---

### Official Review · Reviewer_e5yW · 2022-10-16
**An Interesting Implementation of Video Transformers on the Baby Intuitions Benchmark**

**Rating:** 6
**Confidence:** 3

**Review:**

## Summary
This work implements Video Transformers (VTs) on the Baby Intuitions Benchmark (BIB) and compares performance, particularly on the Violation of Expectations (VoE) measure, across many deep learning-based benchmarks (many RNN-based) as well as a Hierarchical Bayesian (HBToM) approach and real infant data. They used a suite of six different tasks that each test an aspect of common sense reasoning about agent's goals, preferences, and actions. The results show that VTs seem to outperform other neural baselines but not necessarily the HBToM, which is a model that is more carefully curated to exhibit the inductive bias needed to solve this benchmark. The authors did some interesting qualitative analyses (like showing VTs associated with the green door being opened with the end of the episode rather than the key unlocking it) to show some shortcomings of VT.

## Strengths
* Nice economic implementation of VTs by selecting for top-k patches.
* Good blend of quantitative and qualitative methods to show the models' strengths and weaknesses.
* Comprehensive evaluation of various models/benchmarks in this space.

## Suggestions
* I think the qualitative analyses (e.g. layer activations on rational/irrational agents actions, analyzing expectations of the key and door in instrumental actions, etc) are a nice first step towards analyzing the acquired inductive bias towards the model. One thing I think could be interesting to make this systematic is to see if one can do a "probing" analysis on the attention heads of the VT. For example, in Clark et al. 2019 dissected the attention patterns of each head of BERT by seeing if a head's attention patterns can decode various syntactic structures (nouns, prepositions, etc). I'm wondering if a similar approach can be applied to this task - would it be possible to decode the abstract structure of the task from the attention patterns of VT's heads? One could, for example, see if you can decode the parameters of HBToM (agent's preferences, goals, etc) from the attention patterns of VT's.

* Along those lines, this work nicely shows that adding an auxiliary reconstruction loss improves performance. I'm wondering if you could extend this to having auxiliary predictive losses that correspond to things like goals, preferences, etc (parameters that the HBToM explicitly model as mentioned before) ? That may push VTs to rely on the kind of reasoning humans are sensitive to rather than exploiting peculiarities of the training data.

## References
* Clark, K., Khandelwal, U., Levy, O., & Manning, C. D. (2019). What does bert look at? an analysis of bert's attention. arXiv preprint arXiv:1906.04341.

---

### Official Review · Reviewer_j8UB · 2022-10-16
**comparing infants and artificial models on BIB and a qualitative analysis of a transformer-based model’s failure modes**

**Rating:** 6
**Confidence:** 3

**Review:**

quality
- the reasoning for applying a transformer-based method to the BIB dataset is due to previous success of transformers in other domains. If the HBToM method achieves nearly 100% accuracy and mostly outperforms the proposed method, what are the benefits of applying transformers to this domain and comparing its performance with infants?
- Fig7: it appears the transformer-based method’s surprise score is not the most similar to the infant scores among all the models across all tasks, as suggested in lines 25-56. clarified in lines 187-189, this is the case in 2 tasks out of the 6 depicted in Fig7.
- what is the justification for the specific chosen architecture (i.e. the specific modules and layer choice)?
- Table2: useful to provide chance level accuracy and infant accuracy and a definition of model accuracy.

clarity
- missing citations: Kinetics-400, UCF101
- when using the term ‘out-of-distribution’ would be good to specify in what sense (or refer to the Appendix where train vs test is explained)
- Figures 2-4, 6: would be useful to mark clearly what the various entities, starting and end positions are
- typos: line 26 ‘model fails’, 170 ‘have have’, 318 ‘agent. and’

originality
- applying existing architecture (video transformers) to existing benchmark (BIB) with some adjustments for episode length and comparing its performance with infant performance

significance
- a transformer-based model that compares/outperforms other DL baselines on the BIB dataset but is mostly outperformed by Bayesian-based HBToM
- qualitative analysis providing insight into the transformer-based model failure modes
- comparison with infant performance

pros
- DL transformer-based method that is comparable/outperforms other DL baselines on the BIB dataset
- qualitative failure case analysis of the transformer-based method on BIB dataset and comparison with infants

cons
- the transformer-based architecture does not show a systematic quantitative advantage in terms of performance over HBToM in Table2 or in terms of infant behavior similarity over other DL methods in Fig7. In that case, what does the qualitative analysis of the transformer architecture provide?
to show that a DL, non-principled method can be applied in general for inferring goals, preferences and actions similarly to humans, it would be helpful to repeat this analysis in other domains as well.
alternatively, to show DL failure modes compared to humans, a qualitative analysis of the best performing model on each task or the one closest to human scores would potentially shed light on closing the performance gap between AI and infants.